# Dietary-Induced Bacterial Metabolites Reduce Inflammation and Inflammation-Associated Cancer via Vitamin D Pathway

**DOI:** 10.3390/ijms24031864

**Published:** 2023-01-18

**Authors:** Caitlin O’Mahony, Adam Clooney, Siobhan F. Clarke, Mònica Aguilera, Aisling Gavin, Donjete Simnica, Mary Ahern, Aine Fanning, Maurice Stanley, Raul Cabrera Rubio, Elaine Patterson, Tatiana Marques, Rebecca Wall, Aileen Houston, Amr Mahmoud, Michael W. Bennett, Catherine Stanton, Marcus J. Claesson, Paul D. Cotter, Fergus Shanahan, Susan A. Joyce, Silvia Melgar

**Affiliations:** 1APC Microbiome Ireland, University College Cork, T12 YT20 Cork, Ireland; 2School of Microbiology, University College Cork, T12 YT20 Cork, Ireland; 3Teagasc Moorepark Food Research Centre, P61 C996 Fermoy, Ireland; 4School of Biochemistry and Cell Biology, University College Cork, T12 YT20 Cork, Ireland; 5Department of Medicine, University College Cork, T12 EC8P Cork, Ireland; 6National Cancer Screening Service, Cork University Hospital, T12 DC4A Cork, Ireland

**Keywords:** colitis, colitis-associated cancer, high-fat diet, bile acids, vitamin D, proliferation, inflammation

## Abstract

Environmental factors, including westernised diets and alterations to the gut microbiota, are considered risk factors for inflammatory bowel diseases (IBD). The mechanisms underpinning diet-microbiota-host interactions are poorly understood in IBD. We present evidence that feeding a lard-based high-fat (HF) diet can protect mice from developing DSS-induced acute and chronic colitis and colitis-associated cancer (CAC) by significantly reducing tumour burden/incidence, immune cell infiltration, cytokine profile, and cell proliferation. We show that HF protection was associated with increased gut microbial diversity and a significant reduction in Proteobacteria and an increase in Firmicutes and *Clostridium* cluster XIVa abundance. Microbial functionality was modulated in terms of signalling fatty acids and bile acids (BA). Faecal secondary BAs were significantly induced to include moieties that can activate the vitamin D receptor (VDR), a nuclear receptor richly represented in the intestine and colon. Indeed, colonic VDR downstream target genes were upregulated in HF-fed mice and in combinatorial lipid-BAs-treated intestinal HT29 epithelial cells. Collectively, our data indicate that HF diet protects against colitis and CAC risk through gut microbiota and BA metabolites modulating vitamin D targeting pathways. Our data highlights the complex relationship between dietary fat-induced alterations of microbiota-host interactions in IBD/CAC pathophysiology.

## 1. Introduction

The Westernised diet, rich in saturated fat and low fibre content, is associated with a reduction in microbial diversity and low-grade inflammation and is regarded as a risk factor for several metabolic and chronic inflammatory conditions including inflammatory bowel disease (IBD) and cancer [1,2]. The incidence rate of these conditions is increasing worldwide and correlates with the adaptation to a westernised and urban lifestyle, along with a western diet [3]. Although over 200 susceptibility genes are linked to IBD, genetics alone cannot explain this rise suggesting that environmental factors such as diet and the commensal microbiota may be triggering factors [4]. Epidemiological studies have identified a diet rich in saturated fat and low fibre, processed foods, and red meat as risk factors associated with IBD development [1,2]. Chronic IBD, particularly UC, is linked with an increased risk of developing colitis-associated cancer (CAC) [5]. Experimental models of IBD and CAC have generated contradictory results regarding the influence of a high-fat diet and have included limited data on microbiota or metabolite changes [6,7,8,9,10,11,12,13,14]. One seminal study identified *Bilophila wadsworthia*, an opportunistic bacterial pathogen, that bloomed in the presence of taurine-conjugated bile acids or indeed associated taurine. *B. wadsworthia* levels correlated with exacerbated colitis incidence in *IL-10^−/−^* mice fed a fat-saturated milk diet [15]. In contrast, amelioration of inflammation and disease outcomes through consumption of either a high-fat diet or n-6 PUFA diet have been reported for other models of inflammatory disease including acute lung injury endotoxemia and juvenile colitis [16,17]. Overall, it appears that while consuming a high-fat diet may contribute to experimental IBD pathogenesis, the response is variable depending on the experimental model and the mechanisms regulating these responses are unclear. 

The gut commensal microbiota plays a significant role in shaping the intestinal immune system and metabolic status of the host. Concomitantly, the host’s immune system can also influence the composition of the microbiota. An excessive immune response to gut microbiota-associated antigens is one of the inflammatory pathways to tissue damage in IBD [4]. Patients with IBD and CAC have a reduced bacterial diversity, accompanied by increased proportions of Proteobacteria, including adherence-invasive *Escherichia coli* (AIEC) and *Fusobacterium* spp. (especially CAC) [4,18,19,20,21]. Reduced abundance of the anti-inflammatory bacteria *Faecalibacterium prausnitzii*, the butyrate-producing bacteria *Roseburia hominis*, and the secondary bile-acid-modulating *Clostridium* cluster XIVa have been reported in IBD [4,22,23,24]. Bacterial metabolites such as short-chain fatty acids (SCFAs) and bile acids (BAs) have been associated with IBD, with reductions in butyrate, the secondary BAs, lithocholic acid (LCA) and deoxycholic acid (DCA), and in the BA-promoting bacteria *Clostridium* cluster XIVa [23,25]. Receptors binding bile acids include the nuclear receptors farnesoid X receptor (FXR), takeda-G-protein receptor 5 (TGR5), pregnane X receptor (PXR) and vitamin D receptor (VDR). All these BA targets interplay in regulating host metabolism and immune responses and they are generally found on innate immune cells (e.g., macrophages) and non-immune cells [e.g., intestinal epithelial cells (IEC)] [26,27,28]. A focus on BA research in the last few years has identified certain bile acids as immunomodulatory, targeting both the innate and adaptive immune responses, both in in vitro cell systems and in experimental models, indicating a more complex role of BAs in these processes [29,30,31].

The present study aimed to comprehensively dissect the influence of a lard-based high-fat diet on the gut microbiota, microbial function, and their interplay with immune and metabolic responses in experimental models of colitis and CAC. The results indicate a protective effect of a lard-based high-fat diet, that appears to be mediated by bacterial-induced metabolite, LCA. Furthermore, our data show that LCA can ameliorate intestinal inflammation and cancer progression via the VDR and VitD pathway.

## 2. Results

### 2.1. High Fat Feeding Reduces Colitis and Tumour Development

Mice were fed a lard-based control low-fat diet (LF) or HF diet for 5 weeks followed by 5 days DSS and 3 days of water (acute colitis) or 3× DSS cycles (chronic DSS-group) or injected with azoxymethane (AOM) and 3× DSS cycles (colitis-associated cancer, CAC-group) (Appendix A). In agreement with previous reports in C57BL/6 mice, animals with CAC fed a LF diet presented approx. 6.75 tumours/mouse [32], while HF-CAC mice developed significantly fewer tumours, approx. 2.7 tumours/mouse (Figure 1a,b). In addition to the reduced number of tumours, the size of the tumours was, based on visual observations, often smaller in the HF-CAC mice (Figure 1a), however, this was not measured at the time. Histological changes associated with diet treatment in mice with colitis and CAC were examined in haematoxylin and eosin (H&E)-stained colonic sections. The HF diet did not provoke significant histological changes in control animals when compared to the LF-Control (Ctr) group (Figure 1c,d). Acute inflammation was more evident in 67% of the LF-Colitis group compared to 33% in HF-Colitis group. The acute inflammation in the LF-Colitis group was characterized by a mixed inflammatory cell infiltrate consisting predominantly of neutrophils, plasma cells, and histiocytes. The neutrophils were present in the surface epithelium, crypts, and the underlying submucosa and muscularis propria, accompanied by cryptitis and crypt abscess formation (Figure 1e,g). Lymphoid aggregates were also present within the submucosa. Surface ulceration involving at least 2/3 of the crypts and regeneration with crypt depletion was evident in the severely inflamed mice in the LF-Colitis group (Figure 1e,g). In contrast, 33% of the HF-Colitis mice showed focal mild–moderate acute inflammation involving the surface epithelium and predominantly limited to <25% of mucosa (Figure 1f,h). Focal cryptitis and crypt abscesses were still present but in contrast to the LF-Colitis mice, there were no ulcerations or crypt damage (Figure 1e–h). Furthermore, the remaining mice in the HF-CAC group showed low levels of inflammation resulting in a significantly reduced inflammatory score in these mice (Figure 1k). When examining tumour malignancy, the extent of dysplasia and malignancy was more pronounced in the LF-CAC group. All animals in this group showed high-grade dysplasia with 50% of mice suspicious for invasive adenocarcinoma and one mouse displaying frank invasive adenocarcinoma (Figure 1i,j). In the HF-CAC group, 56% of the mice showed high-grade dysplasia. A single HF-CAC mouse showed mild focal acute colitis, whereas the remaining mice presented a low degree of inflammation or malignancy (Figure 1i,j). In summary, the tumour score in the HF-CAC mice was significantly reduced compared to the LF-CAC group (Figure 1k).

### 2.2. High Fat Feeding Reduces Macroscopical Signs of Inflammation

Mice randomly fed a HF diet for four to five weeks showed a gain in approx. 43% body weight compared to approx. 21% in the LF groups (Figure 2). Injection of AOM did not affect weight gain or loss in either LF or HF-fed mice. At W7, HF-fed DSS/CAC-treated mice had significantly less weight loss compared to their LF-fed counterparts, with similar body weight changes in the next 2x DSS cycles (Figure 2). A similar result was observed in mice with acute DSS colitis fed a HF diet (Appendix A). In line with the protection of the HF diet against weight loss, the HF-Colitis and HF-CAC mice had significantly longer and less heavy colons compared to the LF-Colitis and LF-CAC animals, respectively (Appendix A). These unexpected results prompted us to explore whether the order of the diet regimen, i.e., supplied before or after AOM application, affected tumour development (Appendix A). A similar protective effect was noted on clinical markers (body weight) and tumour numbers in animals injected first with AOM followed by HF and cycles of DSS when compared to their LF counterparts (Appendix A), suggesting that the diet is providing the protective effect observed. 

### 2.3. High-Fat Diet Reduces Acute Inflammatory Cell Infiltration and Tumour Malignancy and Systemic and Mucosal Inflammatory Responses

We next examined the impact of the diets on systemic and mucosal inflammatory markers. High-fat diet feeding did not markedly change the plasma cytokine profile compared to the LF-Ctr mice (Figure 3a). Mice with LF-Colitis and LF-CAC showed a significant increase in plasma IL-1β, IFNγ, and IL-12 levels compared to the LF-Ctr mice, which were reduced in the HF-Colitis and HF-CAC groups compared to their LF group counterparts (Figure 3a). An increase in plasma IL-2 and IL-10 levels and a significant reduction in IL-4 levels were noted in the HF-CAC compared to the LF-CAC group (Figure 3a and Appendix A). No significant changes in plasma levels of IL-5, IL-17, TNF-α, and mKC were detected regardless of diet or disease phenotype. Immunophenotyping of splenocytes from mice with HF-CAC showed a significant reduction in F4/80^+^ cells (macrophages) and Ly6G^+^ cells (neutrophils) when compared to the LF-CAC group (Figure 3c), whereas no differences in T, B, or NK cells were observed. A reduction in T cells and neutrophils from mesenteric lymph nodes (MLNs) was seen in the HF-Colitis group compared to the LF-Colitis group, while no major changes in these cell subsets were observed in HF-CAC compared to their LF-counterpart (Appendix A). When we assayed for cytokines in the colon tissue devoid of tumours, no significant differences were seen in the HF-Ctr group when compared to the LF-Ctr group (Appendix A). A reduction in colonic IL-1β, IFNγ, and mKC levels was observed in the HF-Colitis and HF-CAC groups when compared to the LF-Colitis and LF-CAC groups (Appendix A). No significant differences in colonic levels of IL-10 (Appendix A), IL-12p70, and TNF-α were detected regardless of disease or diet treatment. When an in-depth PCR analysis was performed on genes associated with innate cytokines (*Il1β, Il6, Inos*), Th1 (*Ifnγ, Cxcl10*), Th17 (*Il17a*), and Tregs (*Il10, Foxp3*), a general reduction in the expression of these genes was observed in the HF-CAC compared to the LF-CAC mice (Figure 3b). In line with the reduced *Foxp3* gene expression, a significant reduction in colonic Foxp3^+^ cells was observed in the HF-CAC group compared to the LF-CAC group (Appendix A). When markers associated with M1, M2, or metabolic profile were assayed, no major changes in M1 and M2 markers were observed, while an increase in *Sfpr5* and a reduction in *Wnt-5a* was found in HF-CAC compared to LF-CAC mice (Appendix A). In line with the findings in the chronic-DSS model, a reduced inflammatory gene profile was observed in the HF-Acute DSS-colitis group when compared to the LF-Acute DSS-colitis group (Appendix A). 

### 2.4. High-Fat Diet Reduces Proteobacteria and Increases Abundance of Clostridium cluster XIVa in Experimental Colitis and CAC

Next, we examined the impact that diet and the disease state exert on the faecal microbiota composition on samples collected at three different time points, i.e., before DSS start and after AOM injection (W5), during the first cycle of DSS (W7) and immediately prior to culling (W15) in the chronic-DSS and CAC models. A total of 294,347 V4 16S rRNA gene sequence reads were retained after read processing and quality filtering. These corresponded to an average of 49,057 reads per group or 1449 per mouse. Shannon diversity, species richness and coverage estimations were calculated for each data set. Rarefaction curves were seen to be approaching parallel, signifying that extra sampling would yield only a limited increase in species richness. Of the reads, 413,069 (87%) were assigned at the phylum level, 365,218 (76%) at the family level, and 281,785 (59%) at the genus level.

Alpha diversity comparing LF and HF in each of the disease states was conducted using the Chao1 diversity metric. No significant differences (*p* = 0.4594) were observed at W5, as mice were not subjected to DSS or AOM and therefore the diets are compared irrespective of disease (Appendix A). At W7, there was no difference between the LF- and HF-Ctr groups (*p* = 0.6507), whereas a visible but not significant difference between the LF-CAC and HF-CAC groups was seen (*p* = 0.2128). However, a statistically significant difference was observed in the HF-Colitis group compared to the LF-Colitis group (*p* = 0.00563; Appendix A). At W15, a similar but non-significant trend was observed in both the control and the CAC dietary states (*p* = 0.9591 and 0.5490, respectively), while HF-Colitis vs. LF-Colitis was significantly different (*p* = 0.01101; Appendix A). The phylogenetic distance largely agreed with the Chao1 diversity, with only the diets in colitis mice showing a significant difference at W7 (*p* = 0.0001852), whereas at W15 a significance was seen at the 10% level when assessing the diets in both colitis and CAC mice (*p* = 0.08534 and 0.05348, respectively; Appendix A).

Next, the taxonomic composition of the samples was compared between the different groups with Figure 4a displaying the rank of the phylum. At W5 Actinobacteria and Proteobacteria were increased in the LF-Ctr group compared to the HF-Ctr group. Over time, Actinobacteria lost significance but the difference in Proteobacteria remains at each time point throughout the study (Appendix A). At a higher resolution, at W5 the genus *Allobaculum* were increased in the LF-Ctr group, whereas for the *Clostridium* cluster XIVa, *Dorea*, *Oscillibacter* and *Turicibacter* were significantly increased in the HF-Ctr group. At W7, there was just one genus; *Alistipides* differentially increased in the LF group, while at W15 there were no differences (Appendix A). In the colitis groups, a general reduction in Proteobacteria was seen in the HF-fed group throughout the study. Two families were significantly different between the LF- and HF-Colitis groups, Lachnospiraceae and Ruminococcaceae, both of which were increased in the HF group at W7, i.e., during the acute phase of colitis. At the genus level, an increase in the *Clostridium* cluster XIVa was seen in the HF-Colitis group throughout the study (Appendix A). In the CAC groups, at the phylum level, a significant increase in Firmicutes at W5 and W7 and a significantly reduced abundance of Proteobacteria at W15 was found in the HF-CAC group compared to the LF-CAC group. At the genus level, a significant increase in the abundance of *Clostridium* cluster XIVa and *Johnsonella* and a significantly reduced abundance of *Bacteroides* was found in the HF-CAC group compared to the LF-CAC group at W7 (Appendix A).

The beta diversity of the samples based on Bray–Curtis distances and normalisation via the variance stabilising transformation (VST) is shown in Figure 4b. A combination of all time points is shown for each group. The disease states diverge away from the LF-Ctr group along both PC axes 1 and 2. Interestingly, both the HF-Colitis and HF-CAC groups were located closest to the LF-Ctr with the LF-Colitis and LF-CAC groups situated furthest from their relevant controls. To try to find possible drivers of this compositional shift, taxonomic abundances at the genus level were significantly different between diet states and colitis and CAC states were correlated to the PC axes. *Clostridium* cluster XIVa (PC1: −0.25; PC2: −0.059, *p* = 0.0005) correlated in the direction of the Ctr-groups, while *Parabacteroides* (PC1: 0.319; PC2: 0.183, *p* = 8.176 × 10^−9^) and *Bacteroides* (PC1: 0.524; PC2: 0.003, *p* = 2.2 × 10^−16^) shifted in the direction of the colitis/CAC groups. All three of these genera were found to have significantly different abundances between both the colitis and CAC diets. To validate the findings in the chronic-DSS model, the caecal microbiota composition from mice with acute DSS colitis fed LF and HF diets were analysed. An increased abundance of *Clostridium* cluster XIVa and a reduction in *Parabacteroides* and *Bacteroides* were found in the HF-Acute colitis group compared to the LF-Acute colitis group, which is in line with the faecal profile observed in the chronic colitis group at W7 and W15 (Appendix A).

### 2.5. The Quantity of Dietary Fat Influences the Impact of Colitis and CAC on Plasma Metabolic Markers and Tissue Fatty Acids

Next, we examined the metabolic profile in mice fed the diets with colitis/CAC. HF-Ctr mice presented significant increases in % body fat, plasma adiponectin, leptin, insulin, and cholesterol levels and reduced % lean and plasma ghrelin levels compared to the LF-Ctr group (Appendix A). These alterations were significantly reduced in the HF-Colitis and HF-CAC groups compared to the HF-Ctr group, although they were still increased when compared to their LF-Colitis and LF-CAC counterparts. Interestingly these markers were reduced in the LF-Colitis and LF-CAC groups when compared to LF-Ctr. Cholesterol was also reduced in the HF-CAC group compared to the HF-Colitis group (Appendix A).

To address the impact diet and disease have on fatty acids, the fatty acid composition of the liver and short-chain fatty acids (SCFAs) were analysed in the caecum and comparisons were made within each diet group and the disease states. The livers of the LF-Colitis and LF-CAC groups had a significant increase in oleic acid and a reduction in linoleic acid, linolenic acid, γ-linoleic acid, arachidonic acid (AA), eicosapentaenoic acid (EPA), docosahexaenoic acid (DHA), and docosapentaenoic acid (DPA) when compared to the LF-Ctr group (Appendix A). In contrast, increased levels of stearic acid, linoleic acid, linolenic acid, and AA and a reduction in palmitoleic acid and oleic acid levels were detected in the livers of the HF-Colitis and HF-CAC groups when compared to their HF-Ctr counterparts (Appendix A). The livers of mice with HF-CAC presented also significant reductions in myristic acid and palmitic acid when compared to the HF-Colitis and HF-Ctr groups (Appendix A). The data indicate that the liver fatty acids linoleic acid, linolenic acid, and AA, which are negatively affected by colitis or CAC, are improved by a HF diet and disease. When the caecal contents of acetate, propionate, and butyrate were analysed, a significant reduction in acetate was detected in the LF-Colitis group compared to the LF-Ctr group and an increase in acetate levels in the HF-Colitis group compared to the LF-Colitis group (Appendix A). Similarly, caecal levels of propionate and butyrate were significantly increased in the HF-Colitis group compared to the LF-Colitis groups (Appendix A). These data indicate a HF diet increases the concentration of SCFAs in mice with colitis.

### 2.6. High-Fat Diet Induces High Levels of Secondary Bile Acids and Regulates the Expression of Vitamin D Receptor and Vitamin D Target Genes

Findings from microbiota composition analysis that revealed a higher abundance of *Clostridium* cluster XIVa spp in the mice fed a HF diet, instigated us to examine the levels of BAs, since these bacteria are instrumental in the dehydroxylation of secondary BAs [33]. In line with recent findings, a reduction in faecal secondary BAs was seen in mice with CAC [34] and a significant increase in these BAs was detected in all disease groups fed a HF diet when compared to their counterpart groups (Figure 5a) and in total BA (Appendix A). Significant increases in ursodeoxycholic acid (UCDA), DCA, and LCA were identified in the HF-Colitis group compared to the LF-Colitis group (Figure 5a). In addition, primary BAs, UCDA, and DCA were also increased in mice with HF-CAC compared to HF-Ctr, while LCA was significantly reduced in this group when compared to mice with HF-Ctr (Figure 5a and Appendix A). These findings indicated that BAs generated by microbiota may be regulating the recovered phenotype seen in the disease mice fed HF and prompted us to investigate the expression of known receptors and downstream genes recognising secondary BAs, namely TGR5, FXR, and VitD [26,27,28]. We detected a higher mRNA expression of colonic *Tgr5* and *Fxr*, but no difference in the FXR target genes *Ostb* (BA-transporter) and *Fabp6* (BA-importer) in HF-CAC when compared to the LF-CAC counterpart (Appendix A). In contrast, significant alterations in mRNA expression of VitD target genes involved in proliferation and differentiation, apoptosis, and inflammation were seen in the tissue of mice with HF-CAC compared to the LF-CAC groups, represented by a significant increase in *Vdr, Rxrα, Rxrβ, Pxr, p21, E-cadherin, Bax,* and *Caspase-3* expression and a reduction in *CyclinD1, Bcl2, Inos*, and *Il6* expression (Figure 5b,c). A similar gene-expression profile was seen in colons from HF-Acute colitis when compared to their LF-Acute colitis counterpart (Appendix A). These data correlated well with the reduced inflammatory infiltration and recovery of the epithelium observed by histology analysis (Figure 1).

### 2.7. The Secondary Bile Acid LCA Reduces Proliferation and Induces the Expression of Vitamin D Regulated Genes in Lipid-Treated Epithelial Cells 

In line with the reduced gene proliferation profile identified by RT-qPCR (Figure 5c), staining with Ki67 also demonstrated a significant reduction in proliferating Ki67^+^ cells in the crypts of mice with HF-CAC when compared to the LF-CAC group (Figure 6a,b). To further validate the in vivo findings on VitD and BA interaction, we developed an in vitro model whereby the addition of a saturated lipid mixture to the human intestinal epithelial cell line HT29 resulted in proliferation (Appendix A), which corroborates previous reports on murine organoids and obese animals [35]. Notably and in line with the findings in diseased HF-fed mice, the addition of LCA to HT29 cells treated with a saturated lipid mixture resulted in reduced viability and proliferation and an increased expression of VitD-associated genes. A similar reduction in lipid-induced viability was noted in HCT116 cells treated with LCA. Interestingly, the addition of DCA did not show any major impact on these analyses (Figure 6c–e) and an LCA and DCA combination presented a similar profile as LCA-treatment alone.

In summary, a high-fat diet following the induction of colitis and colitis-associated cancer results in a reduction in Proteobacteria and an increase in Firmicutes abundance, especially in *Clostridium* cluster XIVa and in secondary bile acids. Microbiota-derived secondary bile acids, especially LCA, activate the VitD pathway leading to the activation of VitD-regulated genes targeting apoptosis, differentiation, and barrier function and a reduction in genes associated with proliferation and inflammation leading to reduced risk for colitis and colitis-associated cancer development (Figure 7).

## 3. Discussion

In this study, we assessed the mechanism(s) underlying the effect of a lard-based high-fat diet on the gut microbiota, microbial metabolites, and host responses in experimental models of colitis and CAC [32,36]. The results show that a lard-based HF diet reduces experimental DSS-induced colitis and CAC progression, accompanied by reducing epithelial cell proliferation, inflammation, tumour burden, and microbial composition, e.g., Proteobacteria and increases in Firmicutes, especially *Clostridium* cluster XIVa. Furthermore, the secondary bile acid LCA and its receptor VDR and VitD-regulated genes were elevated in mice with ameliorated colitis/CAC and treatment of LCA reduced lipid-induced proliferation and upregulated VitD genes in epithelial cells.

The results are unexpected considering the epidemiological studies reporting saturated fats as risk factors for IBD and previous reports on the worsening of colitis and colon cancer upon high-fat feeding in mice, particularly in APC^Min/+^ mice [7,9,11,13,14,37]. However, our results are consistent with studies in acute colitis and the CAC model [6,12], as well as experimental models of juvenile colitis [17], acute lung injury [16], and chronic social stress [38]. Several factors may account for the apparent differences regarding the contexts and models, including the genetic background (BALB/c, A/J, C57BL/6), the concentration of AOM (8–12.5 mg/kg), the percentage and time of exposure of DSS (2–3%, 5–10 days or cycle DSS), the source of the mice (Charles River, Jackson laboratories, Envigo), and the DSS provider (MP Biochemicals, TdB) [7,9,11,13,14,37]. 

Normal rodent chow is dominated by a high carbohydrate content, contributing to 60–70% energy. In contrast, the carbohydrate content is generally reduced in refined diets such as high-fat diets and contributing to only 20–50% of energy. To validate whether differences in carbohydrate content could have contributed to the discrepancies between our study and previously published studies, we performed an in-depth analysis on these reports. Initially, we noticed that lower carbohydrate energy (19–20%) than our diet (35%) was associated with worsening disease [9,14,37]. However, consumption of a high-fat diet with a higher carbohydrate energy (40–51%) revealed inconsistent disease outcomes as both worsening [11,13] and ameliorating of disease ([6,12], current study) was associated with a higher carbohydrate content. Next, we examined whether the source of fat might also have affected the outcome of disease in the reported studies. Lard-containing high-fat diet was consumed in mice presenting both worsening [9,11,37] and ameliorating disease ([12], current study), with a similar outcome also observed in milk-fat-containing diets [12,13] or in corn-oil-containing diets [6,11]. In summary, differences in diet carbohydrate or source of fat do not entirely explain the differences observed between our study and previous reports. 

When we explored whether the order of the diet regimen, i.e., supplied before or after AOM application affected tumour development, we noted a similar protective effect by a HF diet regardless of order. Notably, feeding a LF diet before AOM was injected appeared to increase susceptibility to AOM as observed by the higher number of tumours observed in these mice (Appendix A). This data further adds another complexity regarding the use of refined control diets in preclinical studies.

A poor diet with saturated fat and low fibre content has long been considered a risk factor for cardiovascular diseases and mortality. However, recent studies have indicated that consumption of products with high sugar content such as beverages, together with saturated fats confers a higher risk for these patients and an increase in mortality [39,40,41]. The intake of sugar-sweetened beverages in adult and adolescent women can also increase the risk of early-onset colorectal cancer [42], which is of relevance in patients with an early IBD onset. Preliminary findings from our group indicated that mice fed a lard-based HF diet containing a higher sucrose content (45KDa/33%) reversed the protective phenotype reported herein with a lower sucrose content (45KDa/17%, unpublished observations). Clinical studies examining the impact of saturated fat and high sugar intake in patients with IBD are rare, although high sugar intake has been identified as a risk factor for its onset [43]. A recent study reported that feeding for a short term an improved standard American diet (iSAD), supplemented with more fruits, vegetables, and fibres but still high in fat and red meat, to UC patients in remission/inactive phase resulted in reduced consumption of refined sugars, improved quality of life, and a modest modulation of the intestinal microbiota and secondary BAs [44]. The authors were unable to determine whether the improvement was due to the consumption of more fibre or less refined sugar, or a change in fat type. Future studies should investigate in depth how the consumption of saturated fats and high sugar can affect IBD pathogenesis.

In agreement with the bacteria profile seen in patients with both IBD and CRC [4,19] and in CAC in C57BL/6 mice [34], a reduction in α-diversity and an increase in Proteobacteria correlated with diseased mice fed a LF diet. HF feeding increased caecal butyrate levels and the abundance of the SCFA-producers Lachnospiraceae and Ruminococcaceae families, which are largely underrepresented in IBD [4,19,45]. At the genus level, an increased abundance of *Dorea, Oscillibacter* and *Clostridium* cluster *XIVa* with reductions in *Bacteroides, Parabacteroides*, and *Turicibacter* was noticed in the HF-fed mice. An increased abundance of *Clostridium* cluster *XIVa*, known to dehydroxylate the primary BAs, CA, and CDCA into secondary BAs in the distal ileum and colon [23,25,33], appears to promote the higher levels of faecal DCA and LCA in mice fed a HF diet with reduced colitis. Contrary to our findings, feeding with a milk-saturated diet to *IL-10^−/−^* mice led to an increase in conjugated bile acids, promoting the blooming of a pathobiont *B. wadsworthia* and aggravating inflammation [15]. These data suggest that certain types of fat can variably influence the microbial environment, bile acid metabolism, and the resulting immune response.

Bile acids can act as signalling molecules, triggering both metabolic and immunoregulatory responses through binding to the nuclear receptors FXR, TGR5, and VDR, as shown in colitis and colon cancer models [27,28]. Patients with IBD are deficient in BA signalling members including a reduced abundance of *Clostridium* cluster XIVa and reduced levels of LCA, DCA and their receptors FXR and VDR [23,45,46,47,48]. A recent study reported that a lack of secondary BAs may promote a pro-inflammatory environment in the gut of UC patients with pouchitis [47], indicating the importance of BA signalling in intestinal homeostasis. In this study, FXR and VDR gene expression was upregulated in the colons of HF-fed mice with a reduced number of tumours. However, neither of the FXR-genes coding for BA importers or transporters were upregulated. Although DCA levels were increased in these mice, no effect by DCA was observed on lipid-induced proliferation or VitD-associated genes in IECs. Other studies have also indicated that DCA treatment can increase tumorigenesis [49,50], potentially due to the activation of ROS, DNA strand breaks, and the suppression of DNA repair [51]. Recent reports have shown that derivatives of secondary BAs, such as 3-oxoDCA, 3-oxoLCA, isoalloLCA, and iso-DCA, can modulate adaptive immune responses by promoting Tregs and reducing Th17 cells [29,31,52,53]. Although Hong and colleagues indicated regulation of Tregs via BAs and VitD [29], a reduction in Foxp3 expression and Foxp3^+^ cells were noted in the HF-protected mice, indicating that Tregs are not the main target for BAs under the current conditions. Feeding a HF diet to colitis/CAC groups led to increased levels of LCA, altered the expression of VDR and VitD-regulated genes, and LCA treatment reduced lipid-induced proliferation with a concomitant induction in VitD-regulated genes in IECs, suggesting that bacterial-produced LCA regulates the reduced immune response seen in the HF-fed animals. Previous studies have reported that VDR acts as a bile-acid sensor for LCA and its tauro-conjugates but does not recognise DCA [54]. Genes encoding several downstream mediators of VitD signalling, including VDR, RXR, and co-repressors, i.e., NCoa and NCoR, and genes associated with BA detoxification (e.g., PXR), and efflux of conjugated BA (e.g., MRP3) [55,56], the latter one previously reported in intestinal slices treated with LCA or VitD 1.25 [57], were induced in the colons of HF-fed mice. Similarly, genes involved in proliferation and differentiation (e.g., Cyclin D1, p21), pro-apoptosis (e.g., Bax, Bak, caspase-3), and inflammation (e.g., iNOS, IL-6); previously shown to be regulated by VitD [58], were altered in HF-fed mice with colitis/CAC. In line with these findings, mice injected with LCA were protected from colitis [52] and epithelial cells treated with LCA resulted in reduced cytokine release [46,52]. Similarly, feeding a highly saturated high-fat diet (24%) or a westernised diet (60%) at the same time as challenging with AOM/DSS reduced CAC progression due to a recovery in goblet cells and mucus production [12]. The impact of LCA or a high-fat diet on the BA profile or the microbiota composition was not investigated in these reports. Overall, these findings indicate that the function(s) of BAs appear to be dependent on several factors including disease type and micro-environment.

In summary, we have identified a protective role of diet-derived microbial metabolites targeting the VitD pathway in regulating inflammation and epithelial cell proliferation in experimental colitis and CAC. The findings highlight the complexity of dietary fat-induced modifications on microbiota–host responses and their relevance in IBD and CAC pathogenesis. Further studies, especially in patients, are needed to mechanistically understand the impact of westernised diets on host–microbe interactions in IBD and CAC pathogenesis.

## 4. Materials and Methods

### 4.1. Mice

Male C57BL/6OlaHsD mice, 5–9 weeks old, weighing 20–22 g, were obtained from Harlan, UK. The mice were housed in an SPF environment (temperature 21 °C, 12 h light: 12 h darkness, humidity 50%). Mice were acclimatised for ≥2 weeks under a control diet with similar content as HF except lard content (Appendix A), herein referred to as a low-fat diet (LF, 10 Kcal% fat, #D12450H, Research Diets, New Brunswick, NJ, USA) before entering the study, and tap water *ad libitum*. Animal husbandry and experimental procedures were approved by the University College Cork Animal Ethics Committee, License B100/4104.

### 4.2. Experimental Design for Colitis and Colitis-Associated Colorectal Cancer and Diet Models

Induction of colitis and colitis-associated cancer (CAC) was performed as described previously with minor changes [32]. The feeding regimen with the diets was adapted from Murphy et al. [59]. After acclimatisation on a LF diet (10 Kcal% fat,), animals were randomly divided into groups of 8–12 mice receiving a high-fat diet (45 Kcal% fat, #D12451, Research Diets) or LF diet (10 Kcal% fat) for 4–5 weeks. Initial studies identified 8 mg/Kg AOM as the best-tolerated dose by the animals in our facility. Mice were injected intraperitoneally (i.p.) with AOM (8 mg/kg, Sigma, Gillingham, UK), rested for 1 week, followed by 3 cycles of 1.5% DSS (w/v, TdB Consultancy, Uppsala, Sweden), each cycle consisting of 5 days DSS and 14 days of water (Appendix A). In a second experiment, mice were injected with 8 mg/kg AOM, followed by randomisation for 4 weeks of a LF and HF diet and 3 cycles of 1.5% DSS (Appendix A). In both setups, mice were sacrificed after the third DSS cycle and 14 days of water (W15, Appendix A). For the acute colitis study, mice were fed LF and HF diets for 5 weeks followed by 2% DSS for 5 days and 4 days of water (Appendix A). All mice were kept under their respective diet during the whole experiment (Appendix A). Food intake, weight, and general health condition were monitored 1–2 times per week as previously described [59]. The composition of the diets is shown in Appendix A.

### 4.3. Collection of Tissue, Fluids, Serum, Caecal and Faecal Content

In Chronic DSS and CAC experiments, faeces were collected from each animal on 3 occasions; (1) before DSS start/1 week after AOM injection (W5); (2) one week post-DSS removal during 1x DSS cycle (W7); and (3) at the end of the study (W15). Caecal content from the acute colitis study was collected at the end of the study. Faeces and caecal contents were used for 16S rRNA sequencing. Fat and lean body mass were measured as described using a Minispec mq benchtop NMR spectrometer (Bruker Instruments, Rheistetten, Germany) [59]. At the end of the experiments, blood (cytokines/metabolic markers), spleen, and MLNs cells (immune phenotyping), liver (free fatty acids, FFA), caecum content (SCFAs), and colon-tumour bearing (tumour+) and non-tumour bearing tissue (tumour-) [RNA, protein and histology, and immunohistochemistry] were collected.

### 4.4. Isolation of Cells from Spleen and MLNs and Flow Cytometry

Cells were isolated from the spleen and MLNs and stained for flow cytometry analysis as previously described [60,61]. Briefly, single-cell suspensions were generated from the spleens and MLNs by mechanical force and isolated cells were stained using antibodies for different cell subsets. Antibodies used in the study are outlined in Appendix A.

### 4.5. Cytokines/Chemokines and Metabolic Markers 

Plasma was assayed for cytokines (mouse TH1/TH2 9-Plex Ultra-Sensitive Kit [IFN-γ, IL-1β, IL-2, IL-4, IL-5, KC/GRO, IL-10, IL-12total, TNF-α, Meso Scale Discovery, MSD technology), adipokines, and metabolic markers [glucose (Fisher Scientific, Infinity Thermo Scientific, Horsham and Loughborough, England), leptin (Millipore, Arklow, Ireland), insulin (Mercodia, Uppsala, Sweden), cholesterol (Infinity Thermo Scientific), ghrelin (Millipore), and adiponection (R&D Systems, Abingdon UK)]. Colon samples were homogenised as described [62], and homogenates were analysed for cytokines using a 7-proinflammatory-plex kit (IFN-γ, TNF-α, IL-1β, IL-6, IL-10, IL-12p70), keratinocyte chemoattractant [mKC] (MSD) and IL-17 (ELISA, R&D Systems). All assays were performed according to the manufacturers’ instructions. Colonic cytokine levels are expressed as pg cytokine/100 mg colonic tissue [62].

### 4.6. RNA Extraction and PCR Analysis

The mRNA expression of colonic tumour+, tumour samples from the animal study, and epithelial cell samples treated with lipid mixture and 2nd bile acids were evaluated by RT-quantitative PCR as described [63]. Briefly, total RNA was isolated using the Absolutely RNA RT-PCR Miniprep kit (Stratagene, Stockport, UK) and Rneasy Mini Kit (Qiagen, Manchester, UK) according to the manufacturer’s protocol. Complementary DNA (cDNA) was synthesized using 1 ug total RNA. Primers and probes were designed using the Universal ProbeLibrary Assay Design Centre (https://www.roche-applied-science.com/sis/rtpcr/upl/adc.jsp, accessed on 16 February 2016). β-actin (beta-actin) was used as housekeeping genes to correct for variability in the initial amount of total RNA. PCR reactions were performed in duplicates or triplicates using 384-well plates on the LightCycler 480 System (Roche, Burgess Hill, UK). Positive and negative controls were included and the 2^−ΔΔCt^ method was used to calculate relative changes in gene expression determined from real-time quantitative PCR experiments [64].

### 4.7. Histology and Immunohistochemistry

Paraffin-embedded tumour^+^ and tumour^−^ distal colons were sectioned at 3 µm and stained with haematoxylin and eosin (H&E), followed by blind scoring by two pathologists (AM and MWB). Inflammation was assessed and graded according to Dieleman et al. 1999 [65]. Tumour scoring is based on the collected scoring of H&E stained sections blindly reviewed by two pathologists (AM and MWB) at 20× magnification. A tumour score from 0 to 5 was assigned according to the following criteria —(0) no sign of cancer; (1) focal low-grade dysplasia; (2) low-grade dysplasia; (3) high-grade dysplasia; (4) intra-mucosal adenocarcinoma; (5) invasive adenocarcinoma. Immunohistology staining for Foxp3 and Ki67 was performed on formalin-fixed, paraffin-embedded colonic tissue sections as described [66]. The number of Foxp3^+^ cells is expressed as the total number of cells per tissue area, and Ki67^+^ cells are expressed as % cells^+^ per crypt. Pictures were taken with an Olympus BX53 widefield microscope with an Olympus DP74 digital camera (Olympus GmbH, Hamburg, Germany).

### 4.8. Epithelial Cell Treatment with Saturated Lipid Mixture and Bile Acids

HT29 cells (passage 10–20) were cultured in complete media until they reached 70–85% confluence, followed by pre-treatment with serum-free medium (untreated control) or lipid mixture (10 to 1% *v*/*v*; Sigma, Gillingham, UK, cat nr L0288) for 24 h, cultured for 24 h with LCA (10 μM) and DCA (50 μM), and assayed for cell proliferation (BrdU Cell Proliferation Assay, Cell Signalling Technology, Stillorgan, Ireland), cell viability (CellTiterGlo, Viability Assay, Promega Corporations, MyBio, Kilkenny, Ireland), and RT-qPCR. The lipid mixture contained non-animal derived fatty acids (2 μg/mL arachidonic and 10 μg/mL each linoleic, linolenic, myristic, oleic, palmitic, and stearic), 0.22 mg/mL cholesterol from New Zealand sheep’s wool, 2.2 mg/mL Tween-80, 70 μg/mL tocopherol acetate, and 100 mg/mL Pluronic F-68 solubilised in cell culture water (Sigma).

### 4.9. DNA Extraction and High-Throughput DNA Sequencing

DNA was extracted from individual faecal and caecal samples using the QIAmp DNA Stool Mini Kit (Qiagen, Crawley, West Sussex, UK) as previously reported [67]. The microbial composition of the faecal samples from the chronic DSS and CAC study was performed according to standard protocols from the manufacturer (Roche Diagnostics Ltd., West Sussex, UK) on a Roche 454 FLX sequencer housed in the Teagasc Sequencing Centre (Moorepark, Fermoy, Cork, Ireland). The microbial composition of the caecal samples collected from the acute-DSS study was performed using the 16S rRNA gene (V3-V4 region) amplified by PCR and the resulting amplicons were sequenced using the Illumina MiSeq platform to generate 2 × 250 pb paired-end reads (Teagasc Moorepark Fermoy, Ireland).

### 4.10. Bioinformatics Sequence Analysis

The V4 region of the 16S rRNA gene was sequenced on a Roche 454 GS FLX for the faecal samples collected at W5, W7, and W15 in the chronic-DSS and CAC study and using the Illumina MiSeq platform for the caecal samples collected at day 9 in the acute-DSS study. The raw read files were processed, and quality filtered in QIIME [68]. OTUs were generated using the USEARCH pipeline [69] and the removal of chimeras (uchime_ref) was carried out using the ChimeraSlayer GOLD database [70]. Sequences were mapped to the OTUs, and a count table was generated, while the taxonomy of each OTU was performed using the mother implementation of the RDP classifier [71] (phylum to genus) and SPINGO [72] (clostridium clusters and species). Alpha and beta diversity was calculated in QIIME and all figures were generated in R. Alpha diversity was calculated using Chao1, observed OTUs, Simpson, and Shannon indices. Beta diversity was calculated using the weighted and unweighted unifrac, and Jaccard and Bray–Curtis dissimilarity. 

### 4.11. Fatty Acid (FAA) Analysis

Lipids from liver samples were extracted and purified according to previous publication by Marques et al. [73]. Briefly, samples were extracted with chloroform: methanol and fatty acid methyl esters (FAME) were prepared and recovered followed by separation by gas–liquid chromatography (Varian 3400; JVA Analytical, Dublin, Ireland) fitted with a flame-ionisation detector (Chrompack, JVA Analytical, Dublin, Ireland) and helium as a carrier gas. The resulting peaks were integrated (Varian Star Chromatography Workstation version 6.0 software) and identification of peaks was done by comparison of retention times with pure FAME standards (Nu-Chek Prep, Elysian, MN, USA). The percentage of each individual FAA was calculated according to the peak areas relative to the total area, with total FAAs set at 100%. The data are shown as means ± SEM in g/100 g FAME.

### 4.12. Short-Chain Fatty Acid (SCFA) Extraction and Analysis

SCFAs were extracted from the caecal contents as previously reported by Marques et al. [73]. Collected caecal samples were vortexed with MilliQ water followed by filtering and transferred to a GC vial and 2-ethylbutyric acid (Sigma). The concentration of SCFAs was determined by gas chromatography (Varian 3500 GC flame-ionization system), a ZB-FFAP column (Phenomenex, Macclesfield, UK). Standards were included in each run to maintain the calibration. Peaks were integrated using Varian Star Chromatography Workstation version 6.0 software.

### 4.13. Bile Acid Extractions

Bile acids were extracted according to Joyce et al. [74]. Briefly, for plasma, 100 μL of plasma was added to 50% ice-cold methanol, spiked with deuterated internal standards. The mixture was vortexed and then centrifuged at 16,000× *g* for 10 min at 4 °C and the supernatant was transferred to fresh tubes for further extraction by the addition of acetonitrile containing formic acid. The supernatant was dried under a vacuum and reconstituted in 50% methanol. For faecal material, 100 mg material was added to Dynabeads (Roche) with 300 μL of ice-cold 50% methanol containing deuterated internal stands of both cholic acid and chenodeoxycholic acid then subjected to five × 30 s rounds of extraction in a dyna-lyser machine (Roche) at 6000 rpm. The mixture was vortexed and then centrifuged for 10 minutes at 10,000× *g* and the supernatant transferred to a fresh tube. An amount of 2 mL of ice-cold acetonitrile with formic acid was added to each tube, vortexed, and agitated at room temperature for 1 h. Samples were centrifuged again to pellet the debris and the supernatant was added to fresh tubes containing 1 mL of ice-cold 100% ACN. The samples were vortexed and dried under a vacuum at 4 °C. The dried extracted bile acids were re-suspended in 150 mL of ice-cold 50% MeOH.

### 4.14. Chemicals for Bile Acid Analysis

The chemicals applied for standard curve construction (organic solvents, bile acids, and their deuterated equivalents) and their providers have been reported previously by Joyce et al. [74]. All standards and internal standard stocks were prepared in water:MeOH (1:1) at 1 mg/mL solution. A list of each analyte examined, and their properties are depicted in Appendix A.

### 4.15. Ultra Performance Liquid Chromatography–Tandem Mass Spectrometry

UPLC-MS was performed using a modified method of Joyce et al. [74]. Briefly, plasma and faecal samples were applied, post-extraction, to a C18 Acquity column (Waters Corp, Wilmslow, UK) and the extracts were eluted analysis using an Acquity UPLC system (Waterrs Ltd, Kettering, UK) and an LCT Premier QTof mass spectrometer (Waters MS Technologies, Ltd, Kettering, UK) Waters software. Masslynx software, specifically Targetlynx, was applied to determine the quality and quantity of each analyte against individual standard curves, whereas Markerlynx was applied for semi-quantitative PCA analysis. 

### 4.16. Statistics

All statistical tests were performed using commercially available statistics software (GraphPad Software, San Diego, CA, USA) unless otherwise stated. Statistical significance was determined with one-way analysis of variance (ANOVA) with post-hoc analysis and two-tail unpaired Student’s *t*-tests. Significant differences in the alpha diversity and taxonomy (at each rank) between groups were tested using a Mann–Whitney test. *p*-values were adjusted for multiple testing using the Benjamini–Hochberg method where appropriate. Data are presented as mean ± SEM unless otherwise stated. A *p*-value of < 0.05 was considered significant. Raw data is available on the SRA under the accession numbers PRJNA552446 and PRJEB53904.

## Figures and Tables

**Figure 1 ijms-24-01864-f001:**
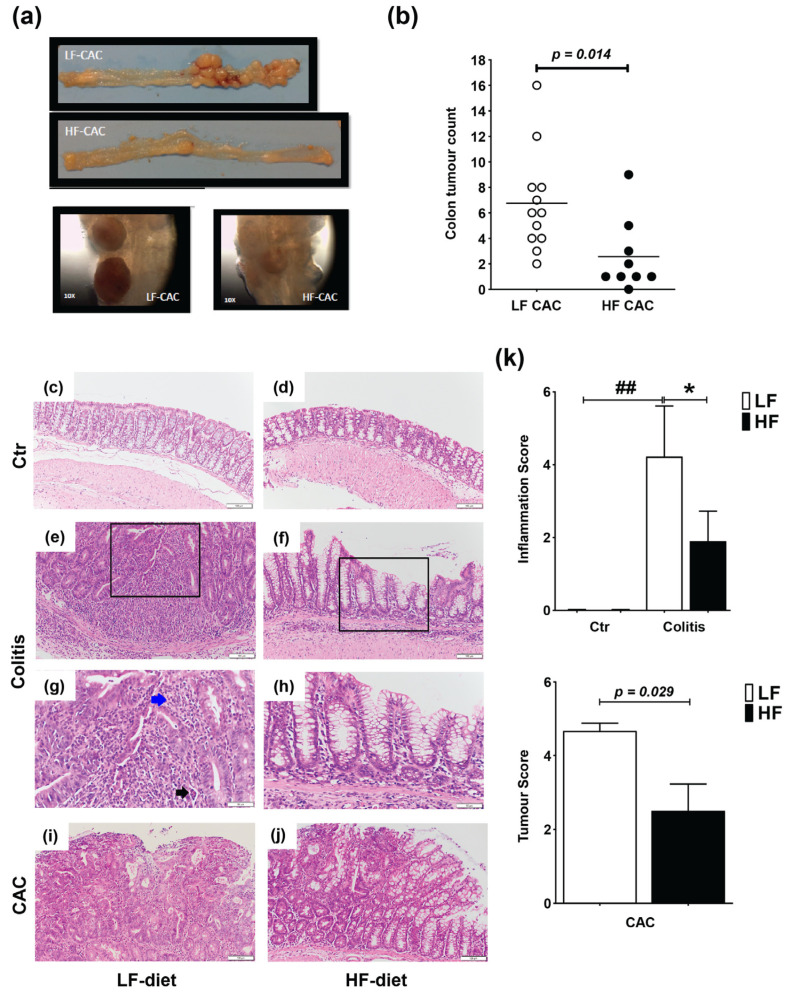
High-fat feeding reduces colitis and tumour development. (**a**) Representative images of the colon with tumours from mice with LF-CAC and HF-CAC at 4× and close-up of tumours, 10×. (**b**) Number of tumours per mouse fed LF and HF diet. Significance determined by two-tail unpaired Student’s *t*-test, * *p* < 0.014. Data are mean ± SEM. n = 8–12 mice per group, representative of two independent experiments. (**c**–**j**) Representative histology tissue sections stained with haematoxylin and eosin of distal colon from (**c**) LF- and (**d**) HF-fed control (Ctr) group; (**e**,**g**) LF- and (**f**,**h**) HF-fed mice with colitis, and (**i**) LF- and (**j**) HF-fed mice with CAC. Pictures in (**c–f**) are taken at 20× zoom (Bar–100 μm) and black boxes are inset in (**g**,**h**), at 40× zoom (Bar–50 μm). Blue arrow indicates neutrophils and black arrow indicates crypt abscess. Pictures in (**i**,**j**) are taken at 20× zoom (Bar–100 μm). (**k**) Histology inflammatory score in LF-/HF-Colitis compared to LF-/HF-Ctr groups, and tumour score in LF-CAC compared to HF-CAC mice. Significances were determined using ANOVA with post-hoc corrections (inflammatory score) and two-tail unpaired Student’s *t*-test (tumour score). * *p* < 0.05 HF-CAC vs. LF-CAC; ## *p* < 0.01 LF-Colitis vs. LF-Ctr. Data are mean ± SEM. n = 6–10 mice per group.

**Figure 2 ijms-24-01864-f002:**
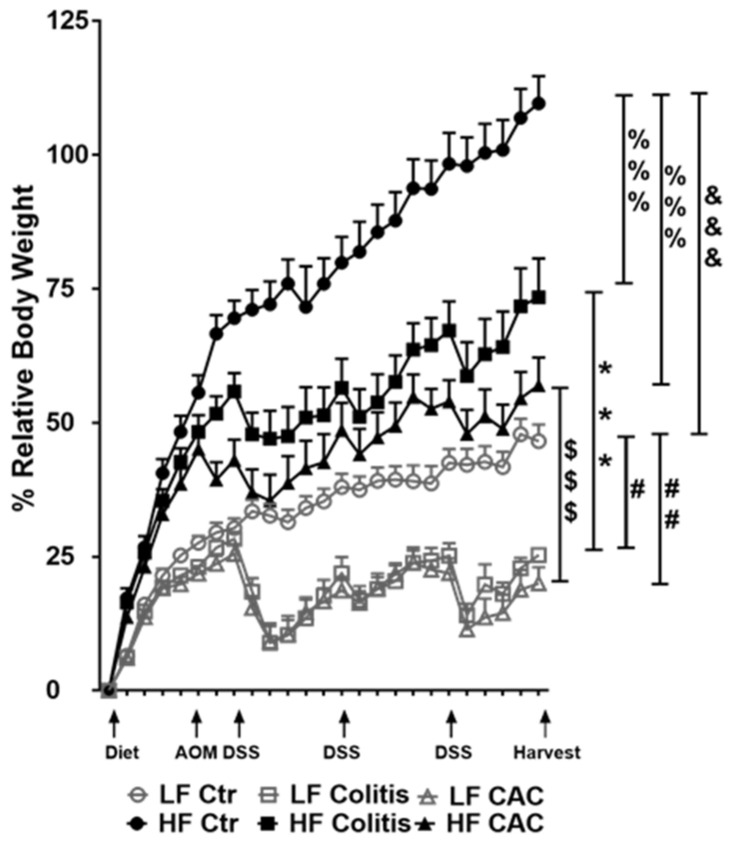
High-fat feeding protects mice from losing body weight upon DSS exposure. Relative body weight change is depicted as a percentage of starting day of diet feeding (day 0). Significances were determined using ANOVA with post-hoc corrections. *** *p* < 0.001 HF-CAC vs. LF-CAC; # *p* < 0.05 and ## *p* < 0.01 LF-Colitis or LF-CAC vs. LF-Ctr; %%% *p* < 0.001 HF-Colitis and HF-CAC vs. HF-Ctr; $$$ *p* < 0.001 HF-Colitis vs. LF-Colitis; &&& *p* < 0.001 HF-Ctr vs. LF-Ctr. Data are mean ± SEM. n = 8–12 mice per group, representative of two independent experiments. Ctr—Control; CAC—colitis-associated cancer.

**Figure 3 ijms-24-01864-f003:**
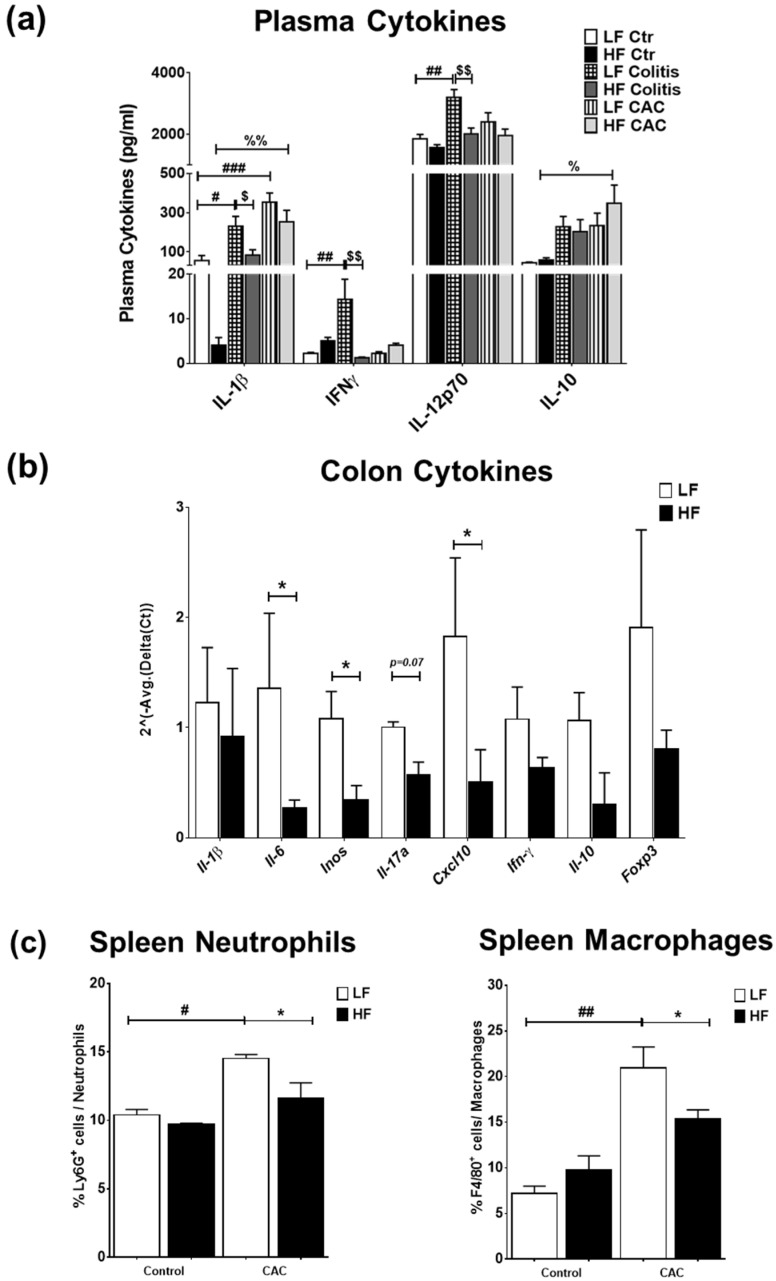
High-fat feeding reduces systemic and mucosal inflammatory markers. (**a**) Plasma levels of IL-1β, IFN-γ, IL-12p70, and IL-10. Data are mean ± SEM. n = 8–12 mice per group, Significances were determined using ANOVA with post-hoc corrections. # *p* < 0.05, ## *p* < 0.01, and ### *p* < 0.01 LF-Colitis vs. LF-Ctr; % *p* < 0.05 and %% *p* < 0.01 HF-CAC vs. HF-Ctr; $ *p* < 0.05 and $$ *p* < 0.01 HF-Colitis vs. LF-Colitis. (**b**) Gene expression of *Il-1β, Il-6, iNOS, Il-17a*, *Cxcl10*, *Ifn-γ*, *Il-10*, and *Foxp3* on tumour-free colonic tissue from mice with CAC fed LF and HF diets, respectively. Expression was determined as n-fold induction compared with the b-actin housekeeping gene and normalised to the LF group. Bars represent the mean ± SEM of three to four mice/group. * *p* < 0.05 HF-CAC vs. LF-CAC as determined by two-tail unpaired Student’s *t*-test. (**c**) Immunophenotyping of splenocytes isolated from LF/HF-fed control and CAC mice. n = 3–12/group. Significances were determined using ANOVA with post-hoc corrections. * *p* < 0.05 HF-CAC vs. LF-CAC; # *p* < 0.05 and ## *p* < 0.01 LF-CAC vs. LF-Ctr. Data is representative of two independent experiments. Ctr—Control; CAC—colitis-associated cancer.

**Figure 4 ijms-24-01864-f004:**
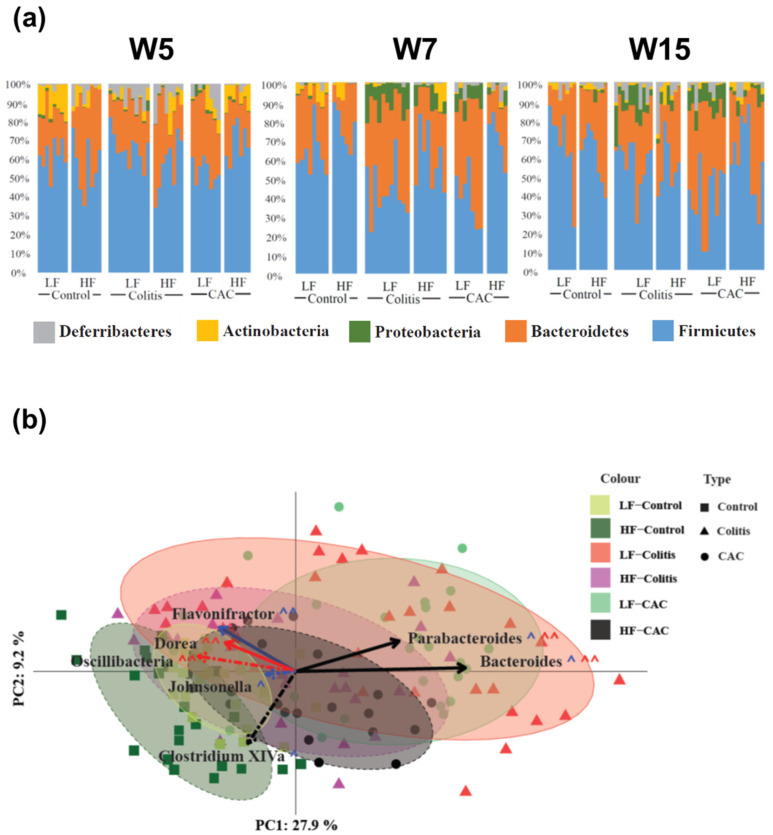
High-fat diet reduces Proteobacteria spp and increases abundance of *Clostridium* cluster XIV in diseased mice. (**a**) Microbial distribution in control, colitis and CAC mice fed LF/HF diet on week 5 (W5, DSS-start and 1 week after AOM-injection), W7 (during the first cycle of DSS) and W15 (at the end of the study). All data presented is at the phylum level. (**b**) Principal coordinates analysis (PCoA) based on Bray–Curtis distance. Ellipses were set to a confidence interval of 80%. Arrows represent Kendall Tau correlations of significantly different genera between the LF-Colitis vs. HF-Colitis and LF-CAC vs. HF-CAC groups. Dashed lines represent an increase in HF, while solid lines show an increase in LF. Black lines show a significant difference in both colitis and CAC disease states, with red being colitis only and blue CAC only. ^ shows significance at the 5% level, ^^ representing the 10% significance level. n = 5–12/group. CAC—colitis-associated cancer.

**Figure 5 ijms-24-01864-f005:**
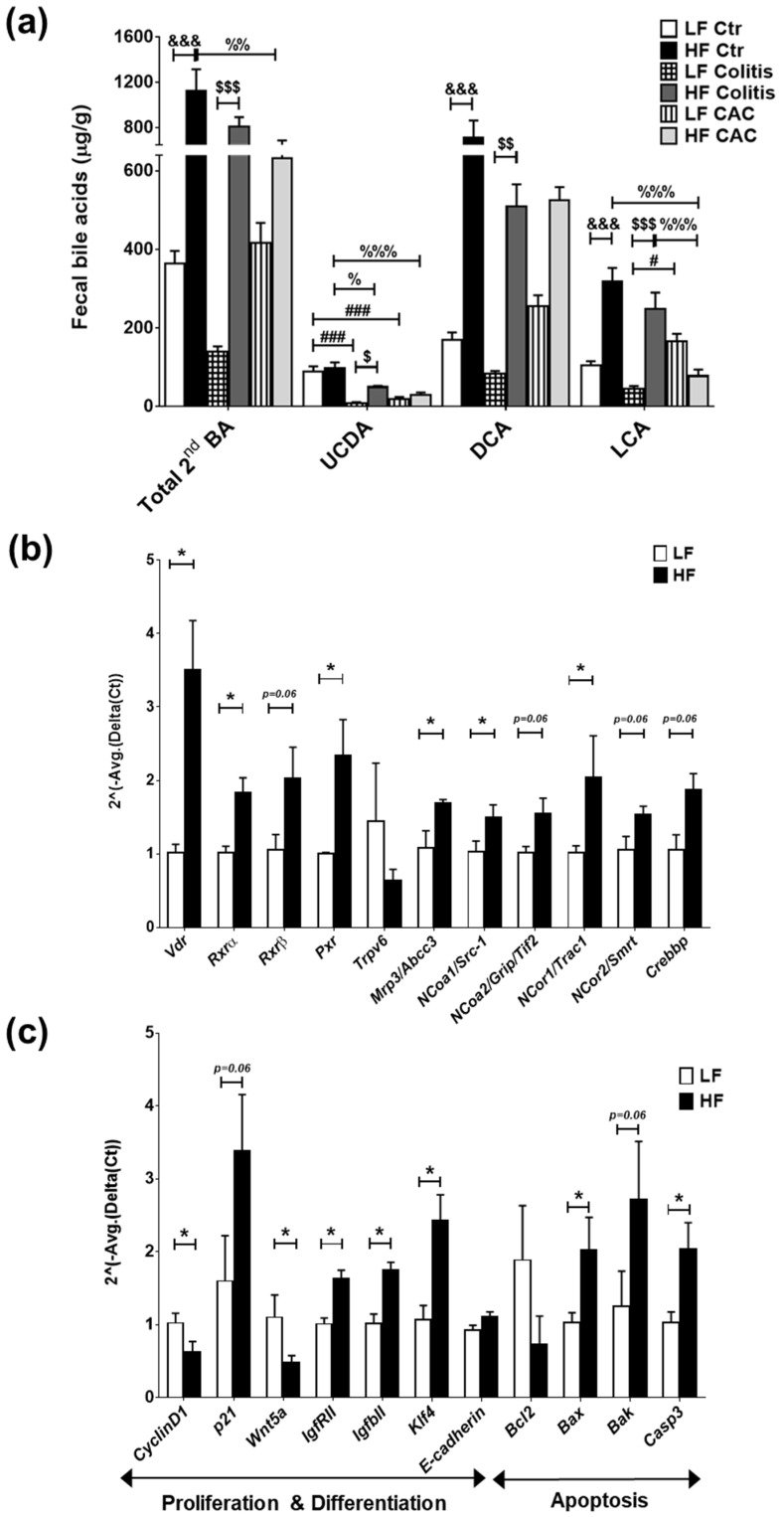
High-fat diet induces LCA which activates VitD-regulated genes in colonic tissue. (**a**) Faecal secondary bile acids UCDA, LCA, and DCA. Data are mean ± SEM. n = 4–5 mice per group. Significances were determined using ANOVA with post-hoc corrections. # *p* < 0.05 and ### *p* < 0.001 LF-Colitis or LF-CAC vs. LF-Ctr; % *p* < 0.05, %% *p* < 0.01, and %%% *p* < 0.001 HF-Colitis or HF-CAC vs. HF-Ctr; $ *p* < 0.05, $$ *p* < 0.01, and $$$ *p* < 0.001 HF-Colitis vs. LF-Colitis; &&& *p* < 0.001 HF-Ctr vs. LF-Ctr. (**b**) Expression of genes in the VitD pathway and (**c**) of VitD-regulated genes in tumour-free colonic tissue from mice with LF-CAC and HF-CAC. Expression was determined as n-fold induction compared with the β-actin housekeeping gene and normalised to the LF group. Bars represent the mean ± SEM, n = 3–4/group. * *p* < 0.05 HF-CAC vs. LF-CAC as determined by two-tail unpaired Student’s *t*-test. Ctr—control; CAC—colitis-associated cancer.

**Figure 6 ijms-24-01864-f006:**
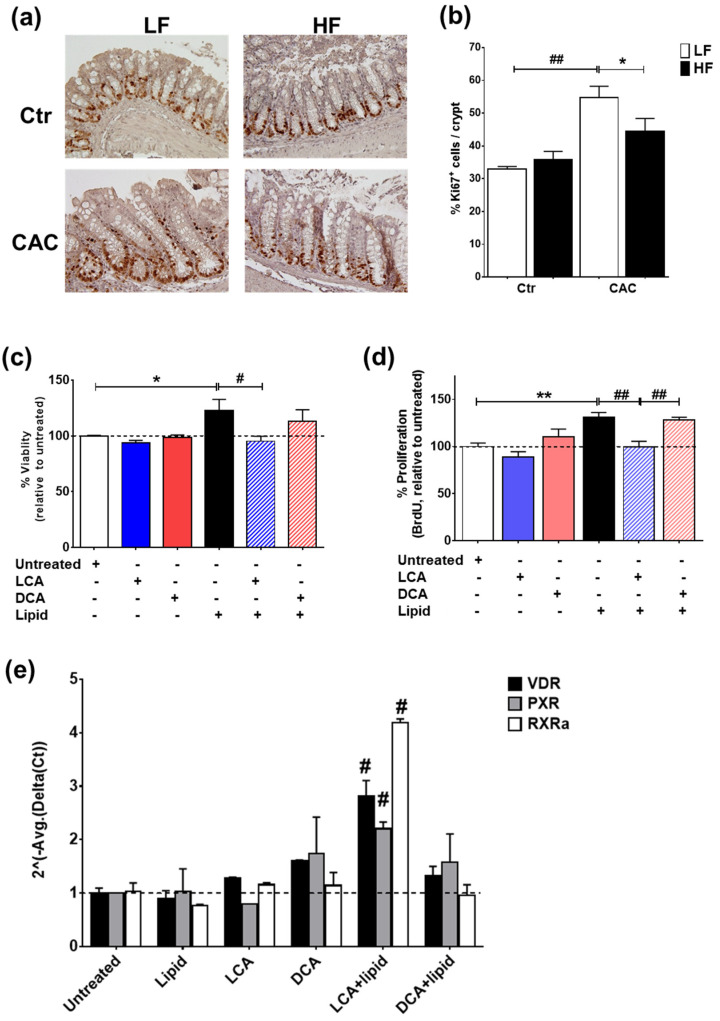
LCA reduces lipid-induced proliferation and induces VitD-regulated genes in intestinal epithelial cells. (**a**) Representative staining of the proliferation marker Ki67 in colonic tissue sections from LF-/HF-Control (Ctr) and LF-/HF-CAC groups. (**b**) Number of Ki67^+^ cells per crypts colonic sections from LF-/HF-Ctr and LF-/HF-CAC groups. n = 4–6 mice/group, 40× magnification. * *p* < 0.05 HF-CAC vs. LF-CAC and ## *p* < 0.01 LF-Colitis vs. LF-Ctr determined using ANOVA with post-hoc corrections. (**c**) Viability, (**d**) proliferation, and (**e**) gene expression analysis of VitD genes VDR, PXR, and RXRa on HT29 intestinal epithelial cells after treatment with saturated lipid mixture (lipid), LCA, DCA, lipid + LCA, and lipid + DCA. Bars represent the mean ± SEM. Representative of two to three independent experiments. Significances were determined by ANOVA with post-hoc corrections (**c**,**d**) and two-tail unpaired Student’s *t*-test in (**e**), * *p* < 0.05 and ** *p* < 0.01 untreated vs. Lipid; # *p* < 0.05 and ## *p* < 0.01 LCA + Lipid vs. Lipid or LCA + Lipid vs. DCA + Lipid.

**Figure 7 ijms-24-01864-f007:**
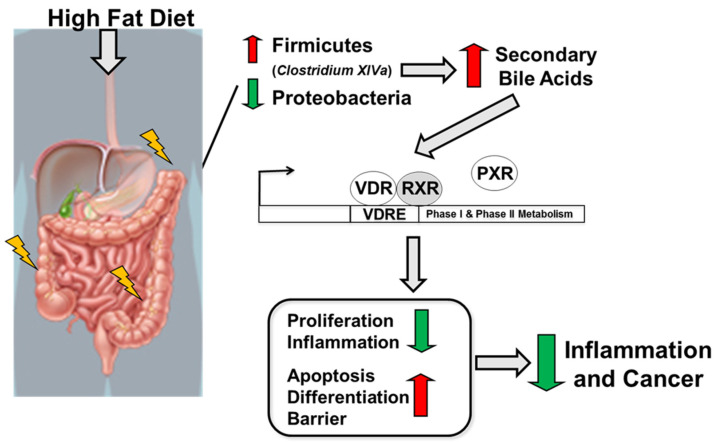
Schematic illustration of the impact of high-fat diet on the host, microbiota, and metabolites in regulating intestinal inflammation and cancer.

## Data Availability

Raw data is available on the SRA under accession numbers PRJNA552446 and PRJEB53904.

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
