# Peer review of "Dietary-Induced Bacterial Metabolites Reduce Inflammation and Inflammation-Associated Cancer via Vitamin D Pathway"

_ijms, 2023, doi:10.3390/ijms24031864_

Round 1

Reviewer 1 Report

The authors undertake an interesting discussion on the complexity of the relationship between the diet, the host microbiome and the influence of inflammatory parameters and the induction of CAC development through the vit. D signaling pathway.

The data from the literature are inconsistent with regard to the advisability of using a high-fat diet, especially in the context of chronic inflammatory diseases and cardiovascular diseases, as the authors emphasize.

It is important that the authors emphasize the low content of carbohydrates in the diet, which undoubtedly has an impact on the obtained results.

The work is written very carefully. The questions that appear while reading are answered later in the work.

Undoubtedly, this is an important voice in the discussion, although further research is required to confirm the thesis.

Author Response

                                                                                    Cork, December 22nd, 2022

Reviewer comments to Manuscript ID: ijms-1822471

Title: Dietary induced bacterial metabolites reduce inflammation and inflammation-associated cancer via Vitamin D pathway

Authors: Caitlin O’Mahony, Adam Clooney, Siobhan F. Clarke, Mònica Aguilera, Aisling Gavin, Donjete Simnica, Mary Ahern, Aine Fanning, Maurice Stanley, Raul Cabrera Rubio, Elaine Patterson, Tatiana Marques, Rebecca Wall, Aileen Houston, Amr Mahmoud, Michael W. Bennett, Catherine Stanton, Marcus J. Claesson, Paul D. Cotter, Fergus Shanahan, Susan A. Joyce, Silvia Melgar

SM – We thank the reviewers and editor for taking the time to critically review our initial submission and for providing constructive feedback and comments to enhance the clarity and quality of our manuscript.

Below you will find a point-by-point response to the queries raised by the reviewers and editorial board.

Amendments are in blue font with my initials (SM) in this response letter and in red font in the revised documents (Main manuscript and Supplementary files). Revised figures will accompany the revised files.

Points from the editorial board

We have checked your manuscript and noticed that the similarity rate with other papers is really high in lines 611-623 and section 4.11, 4.12, 4.14, and 4.15. We would like to ask you to revise this part during major revision.

SM – we apologise for this and have revised the sections and lines as requested.

Reviewer 1

The authors undertake an interesting discussion on the complexity of the relationship between the diet, the host microbiome and the influence of inflammatory parameters and the induction of CAC development through the vit. D signaling pathway.

The data from the literature are inconsistent with regard to the advisability of using a high-fat diet, especially in the context of chronic inflammatory diseases and cardiovascular diseases, as the authors emphasize.

It is important that the authors emphasize the low content of carbohydrates in the diet, which undoubtedly has an impact on the obtained results.

The work is written very carefully. The questions that appear while reading are answered later in the work.

Undoubtedly, this is an important voice in the discussion, although further research is required to confirm the thesis.

SM – We thank reviewer #1 for this positive response and feedback. We have expanded the discussion on how the composition of the diet might have an impact on our study’s outcome and compared to other studies. This is now discussed in lines 438-460.

Reviewer 2

Overall the data is very good, the experiments are thought out and controlled.  I think this paper needs to be reworked and refocused.  The paper needs to be packaged around the diet more. These results are different then other high fat diets, a table looking at the macro nutrients of the diet used in this study compared to other studies should be included so the reader can better understand why these results with high lard high fat diet differ from another high fat diet more western style diet.  The high fat western style diet is associated with obesity, inflammation and increased cancer rates, the focus on the difference in the diet may help to elucidate why these presented results differ from this convention.    

SM – We thank reviewer #2 for the constructive feedback and suggestions. We have revised the discussion around diet, specially focusing on carbohydrate and fat contents as possible explanations to the discrepancies between our study and other published studies (lines 438-460). However, we could not find a clear explanation and instead suggest that is the combination of a number of different factors that affect the outcome of the studies, potentially indicating the difficulty in studying more complex diseases such as IBD and CAC, where the host systems such as the metabolism and chronic immune system are highly responsive. Another aspect to be considered are much of the collected evidence on the impact of westernised diet in humans are generally based on epidemiological correctional data or short-duration feeding trials. Therefore, more studies in patients with these conditions to further dissect the relationship between westernised diets-inflammation-metabolic interactions.

Other points.

  1. The images in figure 1 seem over stained and are difficult to see. Are you certain they are 20x?  they seem lower magnification.  Could higher magnifications be included so that immune cells are visible?

SM – We thank reviewer #2 for this suggestion. A new H&E staining was performed, and new pictures were taken both at 20 and 40x zoom and immune cells have been pointed out with arrows (Figure 1c-j).

  1. Figure 3 B. Is this data HF and LF or HF CAC and LF CAC?  The figure labels differ from the figure legend and the text.  If this is HF CAC and LF CAC the HF and LF should be presented as well for comparison.

SM – We thank the reviewer #2 for pointing this out. We apologise for the error, which has been revised accordingly.

 Reviewer 3

The research utilized preclinical mouse models of inflammatory bowel disease (IBD) and colitis-associated cancer (CAC) to investigate host-microbiota alterations following feeding of a high fat (HF) lard (45%) based diet or a 10% fat diet. The rather unexpected results claiming that a HF lard-based diet protects against IBD and CAC are supported by extensive preclinical data. However, numerous methodological clarifications are warranted to permit full appreciation of the data presented. Specific points for clarification are outlined in “concerns to authors”. In brief, details are needed regarding the diets utilized, the bioassay groups, tumor volume measures, the statistical tests employed, and some aspects of the data are rather incomplete (i.e viability and proliferation in vitro work) or merit further interpretation. Overall, the study presents some new information on the host-microbiota-metabolism axis in the context of dietary fat changes in these preclinical models, but clarification is warranted to more fully appreciate the data and interpret the findings.

SM – We thank reviewer #3 for the constructive feedback to clarify and improve our manuscript.

Concerns to Authors

  1. Methods: An improved diagram and/or table outlining the bioassay in a more comprehensive manner would help clarify the treatment groups and time-line and more clearly depict the actual study groups in the bioassay. The text is challenging to follow in terms of the description alone. A second study is described, but controls remain unclear for example. Considering Figure S1 coupled with a table to clearly define treatment, collections, sample groups and n’s.

SM – We thank reviewer #3 for an excellent comment. We agree that reading through the material and methods can be difficult to assess when and on what samples have been utilised for the different analysis. We have revised the sketches in Figure S1 and have implemented the type of analysis performed on the different samples. Due to space constrains we have not included the number of samples used per analysis, but this information is found under each individual figure legend. We believe this revised figure is more comprehensive for the reader as it provides a global overview of all bioassays performed on the different collected samples.

  1. Methods: The diet needs defined in terms of its complete composition including fat source, carbohydrate, protein and whether it was synthetic, etc. The actual catalog numbers of the diets should be added and tables outlining the diet compositions added to the supplement. This seems key for ascribing the results to the dietary change of a high fat diet alone and not to other elements of the diet. Additionally, a 10% fat diet is not low fat for mice, but really a normal base diet. Lard is rich in linoleic acid which has been linked to increase risk of colitis and colon cancers contrary to these findings.

SM – We thank reviewer #3 for these valid comments. A table containing the composition of the diets have been included (based on information provided by Research Diets) as Supplementary Table 1 and the cat nr of the diets have been amended in lines 546 and 556.  We understand the reviewer’s comments that the diet used as control it is not a low fat diet used in other nutritional studies. However, since we are using a control refined diet, where a major difference is the fat content, we wanted to highlight this by using the term low fat diet. This term has been used by several other reports when referring to a control refined diet (PMID: 36541261; PMID: 36504827; MID: 36436558). We have clarified this point under lines 101 and 544-545. In regards of the rich content of linoleic acid in lard-diet and their role in colitis and colon cancer, although epidemiological studies indicate a risk factor for e.g., ulcerative colitis (PMID: 19628674), preclinical studies have, similar to our study and published reports, generated conflicting results showing both exacerbation or reduction of disease development (PMID: 32594738, PMID: 30279515).

  1. Methods: Line 552, At minimum the imaging modality for body mass determinations should be included.

SM – we thank reviewer #3 for pointing out this and apologise for omitting the information. The imaging instrument used for this measurement is now included in lines 576.

4 4. Methods: Line 578 is unclear. Did you conduct expression studies on both tumor and normal appearing epithelium and if so where was the normal tissue from? And what data shown resulted from this? Relevant data appears in Figure 5, but the legend description is not clear, is b0 from tumor and c) from normal epithelium—unclear.

SM – we thank reviewer #3 for this valid comment. We apologise for the phrasing of this sentence. RNA has been isolated from tissue collected from tumor containing areas (tumor+) and non-tumor containing areas (tumor-). The epithelium reference relates to the intestinal epithelial cell line work. This sentence has been rephrased accordingly “The mRNA expression of colonic tumour+ and tumour-samples from the animal study and of epithelial cell samples treated with lipid mixture and 2nd bile acids was evaluated by RT-quantitative PCR as described”, lines 602-604.

  1. Typo line 597

SM – we apologise for this misspelling. This has been corrected “Ki67”.

  1. Methods: 4.8 Epithelial cell treatment, define the lipid mixture.

SM – we thank reviewer #3 for pointing this out. The content of the commercially available lipid mixture (previously reported in PDMI:26935695) have been added in lines 633 and 636-640.

  1. Methods: Statistics, it is unclear why the Mann Whitney test was utilized for tumor multiplicity data presented in 1b as it is generally not an appropriate statistical test for this determination. This seems a reoccurring theme throughout the paper. Consultation with a statistician may ensure proper test statistics are utilized throughout.

SM – we thank reviewer #3 for the suggestion. After consulting with a statistician, we amended the statistics and figures and figure legends have been amended accordingly.

  1. Methods, details regarding the ‘tumor score’ should be included (i.e. the number of fields scored, at what magnification) and again the Mann Whitney test statistic was applied, but seems an inappropriate test statistic for this measure.

SM – we thank reviewer #3 for the suggestion. The tumour scoring is based on the collected scoring of H&E stained Swiss role sections blindly reviewed by the pathologists (AM and MWB) at 20x magnification. This has been amended in lines 619-620. As mentioned in point 77, after consulting with a statistician we amended the statistics to unpaired t-test.

  1. Tests of significance should be included under the Supplemental Tables/ Figures.

SM – we thank reviewer #3 for the suggestion. This has been amended as recommended under the individual tables and figures.

  1. Supp Table 7 should include a comparison between HF and LF cntr groups as well.

SM – we thank reviewer #3 for the suggestion. The comparison of LF vs HF controls and their significant differences have been added as requested.

  1. Results: Adding quantitative tumor volume information would strengthen the findings given that large tumors can coalesce reducing tumor multiplicity, while expanding the tumor area. The authors do make a size claim on line 107, but quantitative data seems to be missing.

SM – we thank reviewer #3 for an excellent suggestion. Unfortunately, at the time the studies were performed we did not have a Caliper tool to measure tumour volume/size, why the assessment we performed was only visual. We have revised this sentence to reflect that visual assessment rather than Caliper measurements were performed, lines 107-108.

  1. Clarify whether the photomicrographs of H&E sections shown in figure 1 were taken in the same location within the colon.

SM – we thank reviewer #3 for an excellent suggestion. For histology assessment, a longitudinal section of the distal colon is rolled as a Swiss roll and H&E pictures in Figure 1 were taken from the most distal part of the Swiss roll.

  1. Neutrophils are not visible in the H&E described in 1e. Neutrophil specific marker staining would help substantiate this finding.

SM – We thank reviewer #3 for the suggestion. New pictures have been taken from newly stained H&E sections at both at 20 and 40x zoom and presence of neutrophils are depicted by blue arrow as these are stained pink and containing more lobes in the nuclei (Figure 1g). The assessment of the presence of nuclei was performed blindly by two pathologists (AM and MWB).

  1. Cellular images should accompany figure

SM – we are not sure what cellular images and what accompanying figure reviewer #3 is referring to in this comment.

  1. Data presented in Supp 3D is not convinicing.

SM – We thank reviewer #3 for this valid observation and apologise for the low magnification pictures provided in Figure S3D. New pictures have been taken from LF/HF-fed mice with CAC and controls. Foxp3+ individual cells or groups of cells are depicted by black arrows.

  1. Results: Line 348, confusing…..tumor free colons of mice with HF-CAC; how can they be tumor free if they have CAC?

SM – We thank reviewer #3 for this comment. This has been amended accordingly.

  1. Line 496, add “a” before risk factor.

SM – We thank reviewer #3 for this comment. This has been amended as suggested.

  1. Figure 6 Data c) and d) is rather weak in that only a single time point is shown in a single cell line and the viability data does not parallel the proliferation data. At minimum Images should be added and statistical comparisons in the graphic made clear and cell line information added. Finally, were bile acid alone treatments included?

SM – We thank reviewer #3 for these suggestions. Initial kinetic studies were performed when establishing this system which showed that HT29 cells incubated with 5% lipid mixture had a significantly higher viability from 4 hours and onwards (Figure A). However, since HT29 cells need 24 hours under serum-containing conditions to proliferate (PMID: 2684395), and we wanted to examine the impact of lipid mixture and BAs on proliferation, a 24 hours’ culture was chosen for the subsequent experiments. The viability data generated on these studies was based on CellTiterGlo assay, we apologise for the mistake in the material and methods as CellTiterBlue was not the assay used for these studies. This has now been corrected to “CellTiterGlo”. A higher degree of viability was seen in HT29 cells upon culture with lipid mixture, potentially indicating a higher proliferation, which was further validated by staining for Ki67 and by using the BrdU-proliferation assay (Suppl Figure 7a-b). When HT29 cells were cultured under LCA or DCA influence, a reduction in viability was noted and which was corroborated by a reduction in BrdU-proliferation (Figure 6 c and d).

As per the reviewer’s comments, we also validated our findings in a second human intestinal epithelial cell line, namely HCT116, and noticed similar results on viability upon treatment with lipid mixture and after treatment with LCA on lipid treated cells (Figure B). A sentence reflecting these results has been added in lines 386-387. Based on these findings, we believe our data on LCA regulation of lipid-induced epithelial cell proliferation supports the reduced Ki67 staining in HF-fed mice with reduced tumours and colitis. As requested by the reviewer, we have included the effect of the bile acids alone on both cell types as well as clarified the statistical differences on the new figure 6 c and d.

  1. Interestingly Supp Table 6, seems to indicate that most of the effects in terms of tumorigenicity are actually due to the 10% (LF diet) group being sensitive to carcinogen treatments, whereas, the HF groups are rather stable in terms of tumor outcomes regardless of the order of carcinogen treatment. Perhaps this can be worked into the discussion as it seems that the “LF” diet may impart susceptibility to carcinogen treatment versus the HF being protective based on Supp Table 6 data.

SM – We thank reviewer #3 for this valid observation. This has been introduced in the discussion, in line 455-460.

Addition of a new Suppl Table 1 has changed the numbering of the subsequent Suppl Tables. These have been labelled in red font.

As part of the refocus in the discussion on diets, previous lines 494-514 which discusses the impact of sucrose content in high fat diet have been revised and moved forward to lines 461-479. To streamline the discussion other revisions are highlighted in red font.

Changes in reference numbering due to refocused discussion are reflected in red font in the reference list and in the text.

We think that the inclusion of these changes and clarifications to our revised manuscript has improved its overall clarity and quality and we hope it is now acceptable for publication in International Journal of Molecular Sciences.

We thank you for your time and considered opinions and we look forward to hearing back from you soon.

Yours sincerely,

Silvia Melgar, PhD

Host-Microbe and Inflammation Group

APC Microbiome Ireland

Reviewer 2 Report

Overall the data is very good, the experiments are thought out and controlled.  I think this paper needs to be reworked and refocused.  The paper needs to be packaged around the diet more. These results are different then other high fat diets, a table looking at the macro nutrients of the diet used in this study compared to other studies should be included so the reader can better understand why these results with high lard high fat diet differ from another high fat diet more western style diet.  The high fat western style diet is associated with obesity, inflammation and increased cancer rates, the focus on the difference in the diet may help to elucidate why these presented results differ from this convention.     

Other points.

1.        The images in figure 1 seem over stained and are difficult to see.  Are you certain they are 20x?  they seem lower magnification.  Could higher magnifications be included so that immune cells are visible? 

2.       Figure 3 B.  Is this data HF and LF or HF CAC and LF CAC?  The figure labels differ from the figure legend and the text.  If this is HF CAC and LF CAC the HF and LF should be presented as well for comparison. 

Author Response

(The authors gave the same response as above.)

Reviewer 3 Report

Overall:

The research utilized preclinical mouse models of inflammatory bowel disease (IBD) and colitis-associated cancer (CAC) to investigate host-microbiota alterations following feeding of a high fat (HF) lard (45%) based diet or a 10% fat diet. The rather unexpected results claiming that a HF lard-based diet protects against IBD and CAC are supported by extensive preclinical data. However, numerous methodological clarifications are warranted to permit full appreciation of the data presented. Specific points for clarification are outlined in “concerns to authors”. In brief, details are needed regarding the diets utilized, the bioassay groups, tumor volume measures, the statistical tests employed, and some aspects of the data are rather incomplete (i.e viability and proliferation in vitro work) or merit further interpretation. Overall, the study presents some new information on the host-microbiota-metabolism axis in the context of dietary fat changes in these preclinical models, but clarification is warranted to more fully appreciate the data and interpret the findings. 

Concerns to Authors

    1. Methods: An improved diagram and/or table outlining the bioassay in a more comprehensive manner would help clarify the treatment groups and time-line and more clearly depict the actual study groups in the bioassay. The text is challenging to follow in terms of the description alone. A second study is described, but controls remain unclear for example. Considering Figure S1 coupled with a table to clearly define treatment, collections, sample groups and n’s.

2 2. Methods: The diet needs defined in terms of its complete composition including fat source, carbohydrate, protein and whether it was synthetic, etc. The actual catalog numbers of the diets should be added and tables outlining the diet compositions added to the supplement. This seems key for ascribing the results to the dietary change of a high fat diet alone and not to other elements of the diet. Additionally, a 10% fat diet is not low fat for mice, but really a normal base diet. Lard is rich in linoleic acid which has been linked to increase risk of colitis and colon cancers contrary to these findings.

33. Methods: Line 552, At minimum the imaging modality for body mass determinations should be included.

4 4. Methods: Line 578 is unclear. Did you conduct expression studies on both tumor and normal appearing epithelium and if so where was the normal tissue from? And what data shown resulted from this? Relevant data appears in Figure 5, but the legend description is not clear, is b0 from tumor and c) from normal epithelium—unclear.

55. Typo line 597

66. Methods: 4.8 Epithelial cell treatment, define the lipid mixture.

77. Methods: Statistics, it is unclear why the Mann Whitney test was utilized for tumor multiplicity data presented in 1b as it is generally not an appropriate statistical test for this determination. This seems a reoccurring theme throughout the paper. Consultation with a statistician may ensure proper test statistics are utilized throughout.

88. Methods, details regarding the ‘tumor score’ should be included (i.e. the number of fields scored, at what magnification) and again the Mann Whitney test statistic was applied, but seems an inappropriate test statistic for this measure.

99. Tests of significance should be included under the Supplemental Tables/ Figures.

110. Supp Table 7 should include a comparison between HF and LF cntr groups as well.

111.    Results: Adding quantitative tumor volume information would strengthen the findings given that large tumors can coalesce reducing tumor multiplicity, while expanding the tumor area. The authors do make a size claim on line 107, but quantitative data seems to be missing.

112.    Clarify whether the photomicrographs of H&E sections shown in figure 1 were taken in the same location within the colon.

113.    Neutrophils are not visible in the H&E described in 1e. Neutrophil specific marker staining would help substantiate this finding.

114.    Cellular images should accompany figure

115.    Data presented in Supp 3D is not convinicing.

116.    Results: Line 348, confusing…..tumor free colons of mice with HF-CAC; how can they be tumor free if they have CAC?

117.    Line 496, add “a” before risk factor.

118.    Figure 6 Data c) and d) is rather weak in that only a single time point is shown in a single cell line and the viability data does not parallel the proliferation data. At minimum Images should be added and statistical comparisons in the graphic made clear and cell line information added. Finally, were bile acid alone treatments included?

119.    Interestingly Supp Table 6, seems to indicate that most of the effects in terms of tumorigenicity are actually due to the 10% (LF diet) group being sensitive to carcinogen treatments, whereas, the HF groups are rather stable in terms of tumor outcomes regardless of the order of carcinogen treatment. Perhaps this can be worked into the discussion as it seems that the “LF” diet may impart susceptibility to carcinogen treatment versus the HF being protective based on Supp Table 6 data.

Author Response

(The authors gave the same response as above.)
